# CoxBase: an Online Platform for Epidemiological Surveillance, Visualization, Analysis, and Typing of *Coxiella burnetii* Genomic Sequences

Akinyemi M. Fasemore,[a,b,e] Andrea Helbich,[b] Mathias C. Walter,[b] Thomas Dandekar,[c] Gilles Vergnaud,[d] Konrad U. Förstner,[e,g] Dimitrios Frangoulidis[b,f]

[a]University of Würzburg, Würzburg, Germany
[b]Bundeswehr Institute of Microbiology, Munich, Germany
[c]Department of Bioinformatics, Biocenter, Am Hubland, University of Würzburg, Würzburg, Germany
[d]Institute for Integrative Biology of the Cell (I2BC), Université Paris-Saclay, Gif-sur-Yvette, France
[e]ZB MED - Information Centre for Life Science, Cologne, Germany
[f]Bundeswehr Medical Service Headquarters VI-2, Medical Intelligence & Information, Munich, Germany
[g]TH Köln – University of Applied Sciences, Cologne, Germany

**ABSTRACT** Q (query) fever is an infectious zoonotic disease caused by the Gram-negative bacterium *Coxiella burnetii*. Although the disease has been studied for decades, it still represents a threat due to sporadic outbreaks across farms in Europe. The absence of a central platform for *Coxiella* typing data management is an important epidemiological gap that is relevant in the case of an outbreak. To fill this gap, we have designed and implemented an online, open-source, web-based platform called CoxBase (https://coxbase.q-gaps.de). This platform includes a database that holds genotyping information on more than 400 *Coxiella* isolates alongside metadata that annotate them. We have also implemented features for *in silico* genotyping of completely or minimally assembled *Coxiella* sequences using five different typing methods, querying of existing isolates, visualization of isolate geodata via aggregation on a world map, and submission of new isolates. We tested our *in silico* typing method on 50 *Coxiella* genomes downloaded from the RefSeq database, and we successfully genotyped all genomes except for cases where the sequence quality was poor. We identified new spacer sequences using our implementation of the multispacer sequence typing (MST) *in silico* typing method and established *adaA* gene phenotypes for all 50 genomes as well as their plasmid types.

**IMPORTANCE** Q fever is a zoonotic disease that is a source of active epidemiological concern due to its persistent threat to public health. In this project, we have identified areas in the field of *Coxiella* research, especially regarding public health and genomic analysis, where there is an inadequacy of resources to monitor, organize, and analyze genomic data from *C. burnetii*. Subsequently, we have created an open, web-based platform that contains epidemiological information, genome typing functions comprising all the available *Coxiella* typing methods, and tools for isolate data discovery and visualization that could help address the above-mentioned challenges. This is the first platform to combine all disparate genotyping systems for *Coxiella burnetii* as well as metadata assets with tools for genomic comparison and analyses. This platform is a valuable resource for laboratory researchers as well as research epidemiologists interested in investigating the relatedness or dissimilarity among *C. burnetii* strains.

**KEYWORDS** *Coxiella burnetii*, Q fever, genotyping, Web platform, *Coxiella*, typing

Q (query) fever is an infectious zoonotic disease that affects humans and small ruminants like sheep, goats, and cattle. It was first described among abattoir workers in Queensland, Australia, with symptoms of "febrile illness" in 1937 (1). The causative

Address correspondence to Konrad U. Förstner, foerstner@zbmed.de, or Dimitrios Frangoulidis, DimitriosFrangoulidis@bundeswehr.org.

The authors declare no conflict of interest.

agent is a Gram-negative, pleomorphic, obligate intracellular bacterium called *Coxiella burnetii*. It has a worldwide distribution and persists in biological and environmental reservoirs like milk, hay, and dust, which can act as sources for sporadic outbreaks in livestock (2).

Since its first description as a febrile illness in Australia, the pathology of Q fever is now more understood and has been described as usually subclinical in ruminants but may manifest in the form of late-term abortion in pregnant ruminant females (3). In humans, the disease can be observed in two different forms. The first form is acute disease, which is usually self-limiting and might occur alongside symptoms such as febrile illness, fever, and severe headaches. It has been shown to happen in 40% of primary Q fever cases. The second form is the chronic form, usually long-lasting, characterized by endocarditis, and can be severe and, in dire cases, fatal. It occurs in 1 to 5% of primary cases; the remaining cases are usually subclinical/asymptomatic and are also defined as acute disease (2, 4).

The epidemiology of this disease has been linked to the interplay of several dynamic factors, including but not limited to vector diversity, the reservoir type, and the worldwide distribution of the disease (5). Another important point for disease control is the absence of a central platform that connects the different ends of the large and growing field of *Coxiella* research.

As a result, data from *Coxiella* research are dispersed over the academic space and if collected at a point are usually specific to a single method. The implication of these is that the speed of research flow is significantly impeded, especially in urgent cases of outbreaks where strain comparison and discrimination are vital to the control of the etiological agent.

To highlight this challenge, there are up to five known genotyping methods for discriminating *Coxiella* species, namely, multiple-locus variable-number tandem-repeat analysis (MLVA) (6, 7), multispacer typing (MST) (8), IS*1111* typing (9), *adaA* gene typing (10), and plasmid typing (11, 12). MLVA and IS*1111* typing require the measurement of PCR amplification products. MST requires the sequencing of intergenic regions, whereas *adaA* typing is based upon the sequencing of one coding sequence.

All methods allow the detection of a correlation between geographic origin and genotype and are useful for typing strains in regions of endemicity as well as clinical entities (5, 10). The MLVA, MST, and IS*1111* methods offer higher resolution than the other two methods (5).

A researcher interested in typing a new *Coxiella* strain is likely to employ more than a single method to obtain quality proof or at least to employ methods accessible in his particular setting. Access to a database resource with strain information and metadata will be necessary for comparison purposes.

Presently, there are two such resources that house *Coxiella* genotyping data. The first is the MLVA data bank (http://mlva.i2bc.paris-saclay.fr/mlvav4/genotyping/), and the second is the MST database (https://ifr48.timone.univ-mrs.fr/mst/coxiella_burnetii/); for the other genotyping methods, there are no available database resources.

First, we sought to overcome the lack of additional genotyping resources; next, we sought to consolidate the existing resources via the introduction of new features such as the visualization of an allelic reference for MST typing, the aggregation of MLVA groups, and the introduction of MLVA genotypes for better comparison. To this end, we have developed an online, open, web-based platform called CoxBase (https://coxbase.q-gaps.de), which caters to vital aspects of Internet-based *Coxiella* research. This platform also includes a database that contains over 400 *C. burnetii* isolates from different countries. It has been implemented with a user interface for the quick retrieval of isolate information as well as a submission channel to add to the growing body of new *Coxiella* isolates.

Also, we sought to unify all *Coxiella* typing systems under a single platform, alongside all published details of *Coxiella* genotyping, including primers for genotyping protocols, as well as phenotypes, for the purpose of strain discovery and comparison. We implemented an *in silico* genotyping option for all major genotyping systems for *C. burnetii* based on whole genomic sequences.

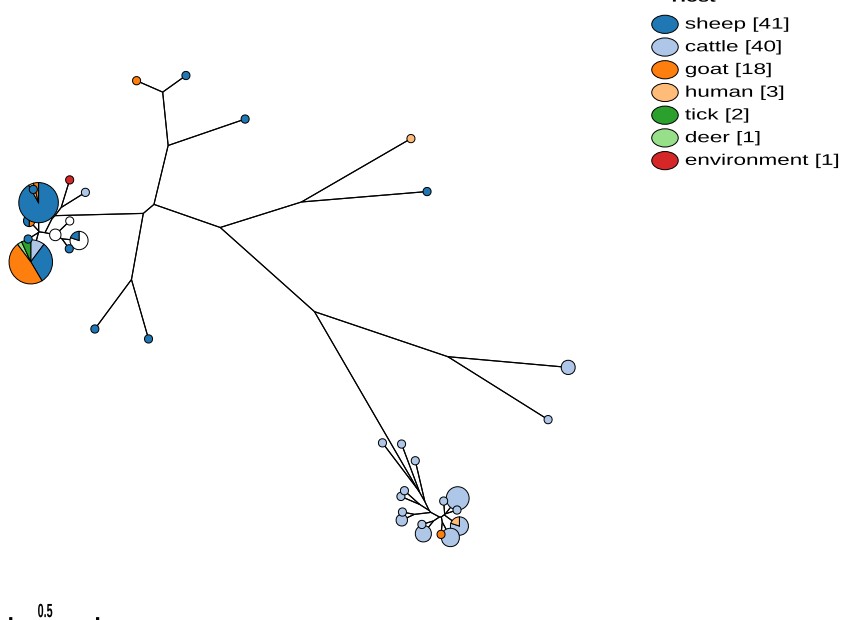

**Host**
- sheep [41]
- cattle [40]
- goat [18]
- human [3]
- tick [2]
- deer [1]
- environment [1]

0.5

**FIG 1** GrapeTree visualization of *C. burnetii* isolates from Germany on CoxBase based on MLVA genotyping. Distinctive clusters based on metadata such as host type can be inferred from such a tree.

Finally, we included visualization systems to quickly summarize all metadata on the country level, maps for the enhanced geographic localization of isolates, and a worldwide distribution map of all *C. burnetii* isolates in our database. Here, we present our platform and its current scope, usage, and capabilities.

## RESULTS

**Genotyping analysis.** We tested our implementation on 50 *Coxiella* genomic sequences from the RefSeq database (see Data Set S1 in the supplemental material). The set contained 11 complete chromosome assemblies, 13 chromosome assemblies, 15 contigs, and 11 scaffolds. The average genome size was 2.01 Mb. The genome sequences in FASTA format were downloaded from the RefSeq database and stored without any modification. The genomes were genotyped individually using the different typing methods on our platform, after which the results were compared to those of known strains in our database. (The results of the implementation test can be found in Data Sets S2 to S4 in the supplemental material.)

**Phylogenetic analysis.** We implemented two types of visualization for phylogenetic trees. The first tree is a GrapeTree (13) implementation that can be used to visualize the genomic relationships of grouped data based on their MLVA profiles (Fig. 1 and 2). The resulting tree can be color-coded based on metadata, is editable, and can also be exported into several image formats. The second tree is implemented using the PhyD3 visualization library (14). This is especially useful for locating MLVA profiles in the MLVA genotype tree, thereby associating a strain with a new MLVA profile with its closest MLVA genotype.

## DISCUSSION

Here, we present a platform that was built with the aim of overcoming the lack of a centralized genomic data resource for *Coxiella burnetii*.

This is the first genotyping platform that combines all the disparate typing systems for *Coxiella burnetii*. Similar platforms exist for other bacterial species, such as PubMLST, albeit usually focused on a single typing system.

Several features are particularly novel and unique: we combined five typing methods to enable the rapid identification of *Coxiella* strains as well as the visualization of the metadata coupled to the geographic distribution. The latter format is particularly

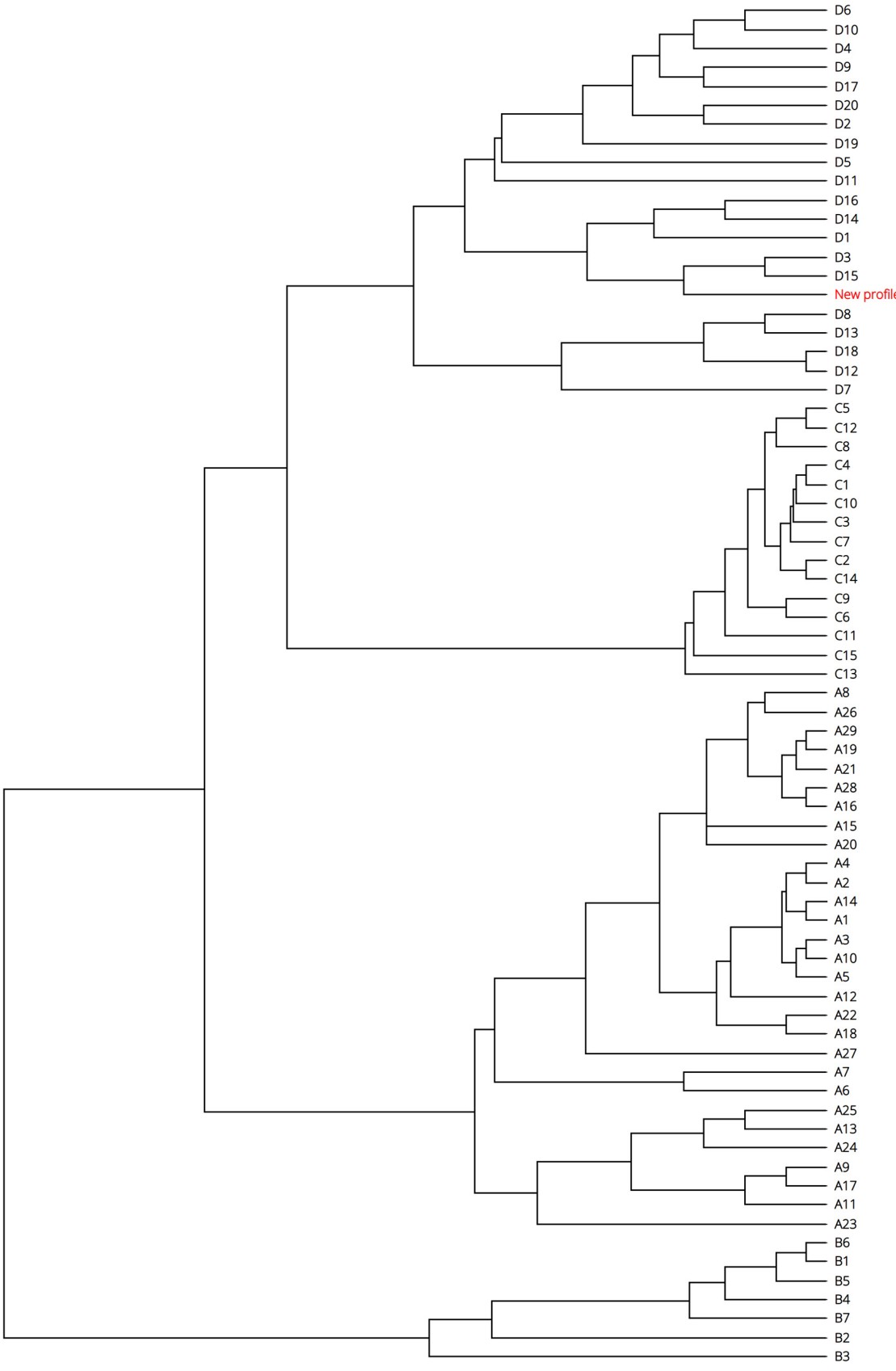

**FIG 2** Unrooted phylogenetic tree of all MLVA genotypes. The highlighted node shows the position of *C. burnetii* strain Q321 that was isolated from cow's milk in Russia. MLVA typing was done via CoxBase.

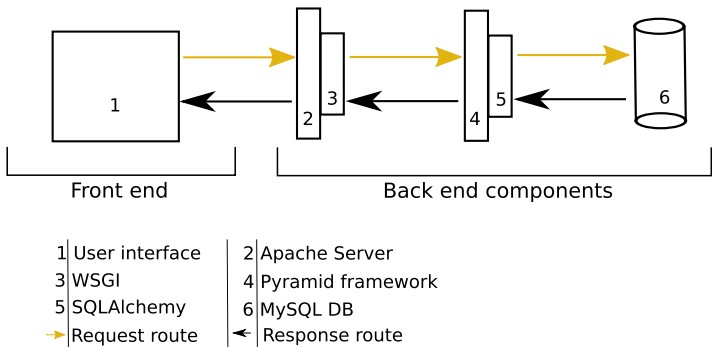

**FIG 3** CoxBase server architecture.

useful to study and control outbreaks, the major shortcoming for which our platform was constructed.

We have also included several features that could assist researchers in understanding the variability within the genomes of *C. burnetii* strains in an epidemiological context. We have leveraged technologies such as next-generation sequencing (NGS), cloud computing, and databases to create an open Web resource that can be used to genotype draft or completely assembled *C. burnetii* genomic sequences as well as compare them to existing strains. Our approach also brought together different aspects of *Coxiella* research, including epidemiological surveillance, sequence analyses, and phylogeny, under a single platform. The strength of *in silico* typing methods relies on, to a significant degree, the quality of the input sequence. Our implementations suggest that *in silico* typing can be an indispensable tool for the rapid genotyping of *Coxiella* genomic sequences. We tested the implementation on 50 *C. burnetii* genomes from the NCBI database, and we were able to type all sequences except for cases where the sequence quality was not good enough. We observed perfect corroboration with known genotypes when we used our implementation to type these sequences, except for one case where we argue that the published profile might not be correct as the observed spacer profile differed in all alleles compared to the published profile. One limitation of our method is in *adaA* gene typing. Although we can distinguish between the different *adaA* gene-positive variants, we are yet to implement a feature to differentiate between the deletion variants (if it is a Q212 deletion or a Q514 deletion). For now, we report only whether the *adaA* gene deletion exists in a given sequence and not the variant of the deletion type. We implemented a retrieval feature on CoxBase that will enable researchers to access the results of their typing analyses up to 3 weeks after their submission date. This would ease collaboration efforts on typing projects and reduce the complexity of information sharing. We have also implemented a genome browser for sequence visualization to accompany sequence typing investigations, most especially primer analysis. Finally, we implemented a submission feature for researchers who wish to share new MLVA or MST profiles. We hope that this platform will provide researchers with the opportunity to investigate the variability among *C. burnetii* genomes as well as help to better understand the epidemiology of Q fever disease in terms of genotype correlations with metadata like host specificity and geographic information. We will update the platform periodically to keep the data current and curated.

## MATERIALS AND METHODS

The systems architecture of CoxBase is described in Fig. 3. It consists of the following components.

**Web server components.** The server is run by an Apache HTTP server on a machine hosted by de.NBI Cloud services. The server components can be grouped under 2 main sections, the front end and the back end.

**TABLE 1** MLVA markers used and their primer sequences

| Marker | Primer sense | Primer sequence |
|--------|--------------|-----------------|
| ms01 | Forward | GCCCTTGTCATCTTGCGG |
|  | Reverse | TCAAGTATTAATGAGCGTCG |
| ms03 | Forward | TGTCGATAAATCGGGAAACTT |
|  | Reverse | ACTGGGAAAAGGAGAAAAAGA |
| ms20 | Forward | CTGAAACCAGTCTTCCCTCAAC |
|  | Reverse | CTTTATCTTGGCCTCGCCCTTC |
| ms21 | Forward | AGCATCTGCCTTCTCAAGTTTC |
|  | Reverse | TGGGAGGTAGAAGAAAAGATGG |
| ms22 | Forward | GGGGTTTGAACATAGCAATACC |
|  | Reverse | CAATATCTCTTTCTCCCGCATT |
| ms23 | Forward | GGACAAAAATCAATAGCCCGTA |
|  | Reverse | GAAAACAGAGTTGTGTGGCTTC |
| ms24 | Forward | ATGAAGAAAGGATGGAGGGACT |
|  | Reverse | GATAGCCTGGACAGAGGACAGT |
| ms26 | Forward | GCAATCCAGTTGGAAAGAA |
|  | Reverse | ATTGAAGTAATCCATCGATGATT |
| ms27 | Forward | TTTTGAGTAAAGGCAACCCAAT |
|  | Reverse | CAAACGTCGCACTAACTCTACG |
| ms28 | Forward | AATGGAGTTTGTTAGCAAAGAAA |
|  | Reverse | AAAGACAAGCAAAACGATAAAAA |
| ms30 | Forward | ATTTCCTCGACATCAACGTCTT |
|  | Reverse | AGTCGATTTGGAAACGGATAAA |
| ms31 | Forward | ACAGGCCGGTATTCTAACC |
|  | Reverse | CCTCAGCACCCATTCAG |
| ms33 | Forward | TAGGCAGAGGACAGAGGACAGT |
|  | Reverse | ATGGATTTAGCCAGCGATAAAA |
| ms34 | Forward | TGACTATCAGCGACTCGAAGAA |
|  | Reverse | TCGTGCGTTAGTGTGCTTATCT |

**(i) The front end.** The main component of the front end is the Web user interface; this is designed to accept user queries as well as submissions, send data to the back end, and present data back to the user. Styling was achieved through an assortment of the cascading style sheet (CSS) Bootstrap framework (https://getbootstrap.com/), the jQuery UI library, and custom CSS style scripts. The validation of form and event processing is achieved with JavaScript. The user interface accepts two kinds of data input: FASTA-formatted whole genomic sequences (contigs or complete assemblies) for typing purposes and typing profiles via multiple-locus variable-number tandem-repeat analysis (MLVA) and multispacer sequence typing (MST) for isolate comparison and discovery.

**(ii) The back end.** The back end handles user requests and uses a MySQL database to store data. Requests are handled via an Apache server (https://httpd.apache.org/), which then communicates via the Web Server Gateway Interface (WSGI) to a Python pyramid framework application (https://trypyramid.com/). The application processes the request and communicates via the SQLAlchemy library (https://www.sqlalchemy.org) to the MySQL storage.

**Genome typing.** We have implemented five different *in silico* typing methods for *Coxiella* sequences on the server: the MLVA typing method (6), the MST method (8), the *adaA* gene typing method (10), the plasmid typing method (12), and the IS*1111* typing method (9). The typing programs were implemented in the Python Web application.

**Establishing the typing features. (i) MLVA typing.** The MLVA typing feature accepts as the input genomic sequences either as contigs or as a complete assembly in FASTA format. The lengths of 14 MLVA amplicons (when present) are extracted *in silico* with the e-PCR tool (15) using primers described

| IS element | state | IS element | state | IS element | state | IS element | state | IS element | state |
|---|---|---|---|---|---|---|---|---|---|
| IS1111-1 | + | IS1111-2 | + | IS1111-3 | + | IS1111-4 | + | IS1111-5 | + |
| IS1111-6 | + | IS1111-7 | + | IS1111-8 | + | IS1111-9 | + | IS1111-10 | + |
| IS1111-11 | + | IS1111-12 | + | IS1111-13 | + | IS1111-14 | + | IS1111-15 | + |
| IS1111-16 | + | IS1111-17 | + | IS1111-18 | + | IS1111-19 | + | IS1111-20 | + |
| IS1111-21 | - | IS1111-22 | - | IS1111-23 | - | IS1111-24 | - | IS1111-25 | - |
| IS1111-26 | - | IS1111-27 | - | IS1111-28 | - | IS1111-29 | - | IS1111-30 | - |
| IS1111-30 | - | IS1111-31 | - | IS1111-32 | - | IS1111-34 | - | IS1111-35 | - |
| IS1111-36 | - | IS1111-37 | - | IS1111-38 | - | IS1111-39 | - | IS1111-40 | - |
| IS1111-41 | - | IS1111-42 | - | IS1111-43 | - | IS1111-44 | - | IS1111-45 | - |
| IS1111-46 | - | IS1111-47 | - | IS1111-48 | - | IS1111-49 | - | IS1111-50 | - |
| IS1111-51 | - | IS1111-53 | - | IS1111-54 | - | IS1111-55 | - | IS1111-56 | - |
| IS1111-57 | - | IS1111-58 | - | IS1111-59 | - | IS1111-60 | - | IS1111-61 | - |
| IS1111-84 | - | | | | | | | | |

**FIG 4** IS*1111* typing results for RSA 439 as calculated on the CoxBase platform. IS, insertion sequence.

previously by Frangoulidis et al. (Table 1) (7) (updated at http://mlva.i2bc.paris-saclay.fr/MLVAnet/spip.php?rubrique50). The repeat number is calculated with the following formula:

$$RN = \frac{(AL - FL)}{RS}$$

where RN is the repeat number, AL is the amplicon length, FL is the flanking length, and RS is the repeat size.

There are **24** isolate(s) with this MLVA Genotype

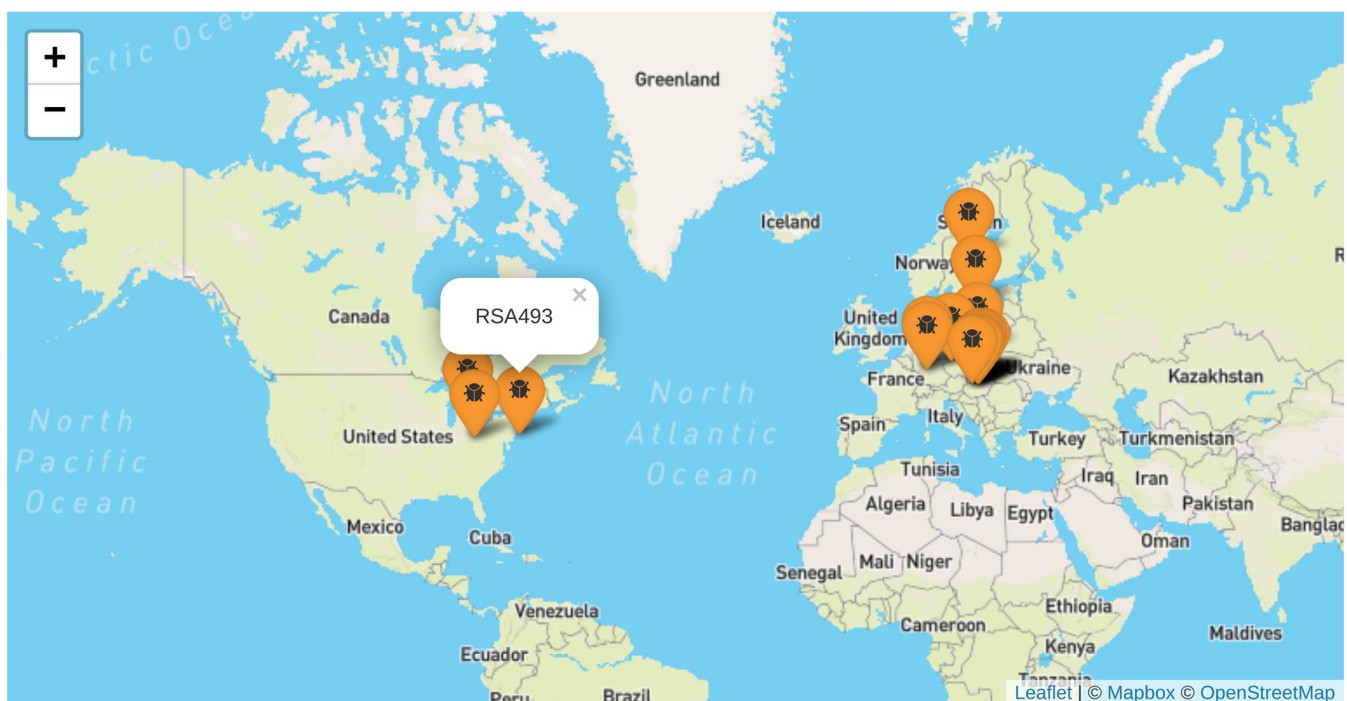

**FIG 5** Geographic visualization of the locations of isolates belonging to the B1 MLVA group as seen on CoxBase.

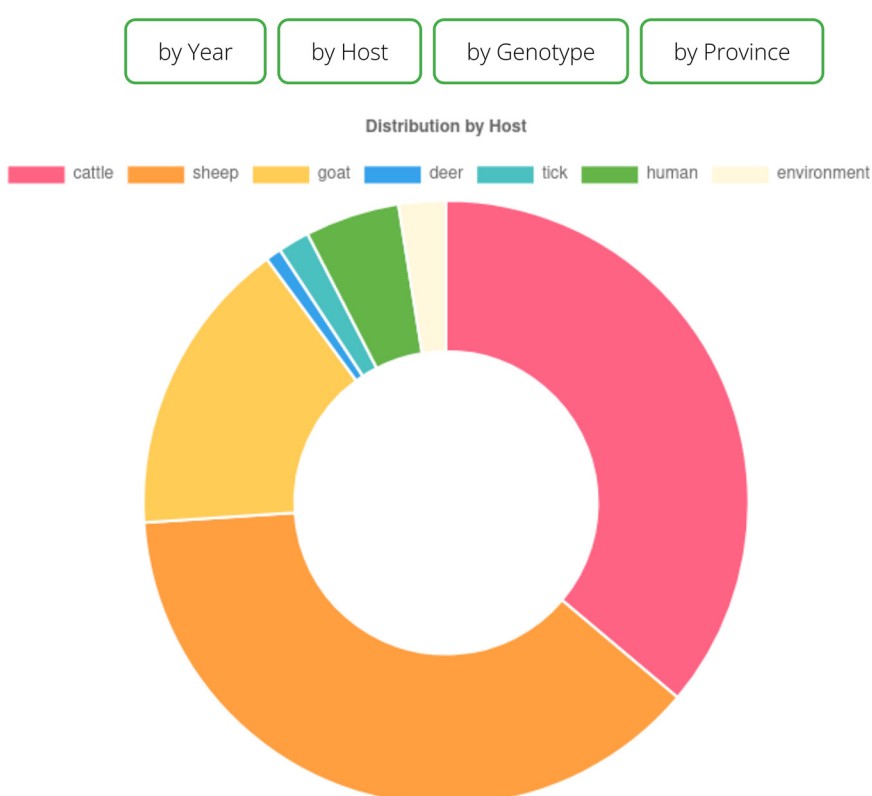

## Distribution Charts

| by Year | by Host | by Genotype | by Province |

**Distribution by Host**

■ cattle  ■ sheep  ■ goat  ■ deer  ■ tick  ■ human  □ environment

**FIG 6** Donut plot of host data from Germany showing that the most common hosts are sheep and cattle.

For every submitted job, a unique identifier is generated that can be used to retrieve the results historically from the database within 3 weeks after the date of submission. The results of MLVA typing are presented in the form of a table with all the calculated parameters. A feature to search the database for closely related MLVA profiles is also provided.

**(ii) MST.** The *in silico* MST method accepts genomic sequences in FASTA format. The first step is amplicon detection via USEARCH (16). This is done using the MST primers described previously by Glazunova et al. (8). The allele type is determined by aligning the detected amplicon sequence globally with known alleles in the MST library (https://ifr48.timone.univ-mrs.fr/mst/coxiella_burnetii/spacers.html). Novel sequences with no match are also reported. The detected MST profile can be used as a query to the database to find the corresponding MST group.

**(iii) IS*1111* typing.** IS*1111* typing is based on the detection of localizations adjacent to IS*1111* elements (9). This is a binary detection method, meaning that discrimination is based on the absence or presence of an amplicon in a given location. For *in silico* detection, we employed the e-PCR tool (15) to detect amplicons based on primers described previously (9) and extended by P. Bleichert and M. Hanczaruk (unpublished data). Presence or absence is highlighted with green or red, respectively, as shown in Fig. 4.

**(iv) *adaA* and plasmid typing.** The *adaA* phenotype was previously reported to correlate with plasmid type (10); therefore, we combined these two typing methods. Five different variants of the *adaA* gene have been reported, three single nucleotide variants (wild type, A431T single nucleotide polymorphism [SNP], and repeat) and two deletion variants (Q154 deletion and Q212 deletion) (10). In our implementation, we first try to detect if the coding sequence of the *adaA* gene exists within the genome to be typed. For this, we used the USEARCH tool (16) and the primer sequence for the detection of the entire *adaA* open reading frame (684 bases) (10). If an amplicon exits, we subsequently evaluate its length. If the length is longer than 684 bases, we assign it the *adaA* insertion genotype, and if it is shorter, we assign it the incomplete *adaA* genotype. If it is exactly 684 bases, we evaluate the type of SNP at position 431 of the amplicon sequence. For the detection of the plasmid type, we employed 4 primers that have been used for the direct identification of *C. burnetii* plasmids via laboratory PCR methods (11, 12, 17).

**Isolate discovery and comparison.** The CoxBase platform offers features for the discovery and comparison of *Coxiella* strains through several approaches. One approach is to query the database based on metadata and genotype features like country, host type, plasmid type, year of isolation, MLVA genotype, and MST group. The advantage of this approach is that it is fine-grained, and the fields can be aggregated to build more specific queries. Another approach utilizes a faceted search; this approach is more suitable for refining queries based on reviewed criteria. Other approaches rely on making queries based on known typing profiles via MLVA or MST schema. This is implemented as follows.

For users who wish to discover isolates with a specific isolate profile (MST or MLVA), they need to provide a complete or partial profile (MLVA or MST) of the isolate that they are interested in. Usually, one marker is enough for a search, but for more defined and reliable results, at least 6 markers should be provided for the MLVA query, and 10 should be provided for the MST query. For ease of comparison, isolates with similar profiles are pooled in a single row in the query results. Profile entries can then be expanded with the click of a button called "View profile entries" in the final column of the result table. A list of all isolates with that profile is provided with metadata. Geographic information is visualized through a Leaflet map (https://leafletjs.com/). The aim is to provide a geographic orientation that can be used to estimate the physical proximity of the isolates. Figure 5 shows the geographic visualization of the locations of isolates from the B1 MLVA group; the historical strain RSA 493 is highlighted. The last approach relies on grouping based on geographic location. A user interested in isolates from a particular country will approach the distribution map. A comprehensive table of all isolates from the country of interest can be seen after clicking on the country marker located on the map.

**Visualization.** We implemented an interactive visualization feature based on the Chart.js (https://www.chartjs.org/) JavaScript visualization library. This can be accessed through the dashboard link on country markers in the distribution map. Distribution plots for metadata categories such as host type, year of isolation, place of isolation, as well as genotype could help answer questions such as the most predominant host type in a particular location, as illustrated in Fig. 6.

**Data availability.** The source code for this project has been deposited at GitHub (https://github.com/foerstner-lab/CoxBase-Webapp). The platform was developed as part of the Q-GAPS consortium, and due to the occasionally sporadic outbreaks of Q fever, the project partners have the need to keep the resource available and updated. Hence, it will be updated with user submissions after curation on a monthly basis.

## SUPPLEMENTAL MATERIAL

Supplemental material is available online only.
**DATA SET S1**, CSV file, 0.01 MB.
**DATA SET S2**, CSV file, 0.01 MB.
**DATA SET S3**, CSV file, 0.01 MB.
**DATA SET S4**, CSV file, 0.01 MB.

## ACKNOWLEDGMENTS

This research has been funded by the Federal Ministry of Education and Research of Germany (BMBF) (project number 01KI1726E). Furthermore, the work was supported by the BMBF-funded de.NBI Cloud within the German Network for Bioinformatics Infrastructure (de.NBI) (031A537B, 031A533A, 031A538A, 031A533B, 031A535A, 031A537C, 031A534A, and 031A532B).

We declare no competing interests.

A.M.F., M.C.W., and K.U.F. wrote the software and the database; A.H. organized the project; A.M.F. prepared the manuscript; D.F. and G.V. conceived the idea of the project; K.U.F., D.F., and T.D. supervised the project; and all authors reviewed the manuscript.

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
