## [Reviewer comments · mSystems]

CoxBase: an online platform for epidemiological surveillance, visualization, analysis and typing of *Coxiella burnetii* genomic sequences.

Akinyemi Fasemore, Andrea Helbich, Mathias Walter, Gilles Vergnaud, Thomas Dandekar, Konrad Förstner, and Dimitrios Frangoulidis

Corresponding Author(s): Konrad Förstner, ZB MED - Information Centre for Life Science

Review Timeline:

Submission Date:	April 1, 2021
Editorial Decision:	August 20, 2021
Revision Received:	October 26, 2021
Accepted:	October 27, 2021

Editor: Jonathan Eisen

Reviewer(s): The reviewers have opted to remain anonymous.

Transaction Report:

DOI: <https://doi.org/10.1128/mSystems.00403-21>

August 20, 2021

Dr. Konrad U Förstner
ZB MED - Information Centre for Life Science
Data Science and Services
Cologne
Germany

Re: mSystems00403-21 (CoxBase: an online platform for epidemiological surveillance, visualization, analysis and typing of *Coxiella burnetii* genomic sequences.)

Dear Dr. Konrad U Förstner:

Thank you for submitting your manuscript to mSystems.

With sincere apologies, this manuscript has just taken a really long time to get reviews. Multiple people agreed and then did not provide reviews in time. We got one review which believes this manuscript can be of use but expresses some concerns with presenting new results in an article that is really focused on a web resource / database. I believe that minor modifications could address the concerns raised. Of course, acceptance is not guaranteed until you have adequately addressed the reviewer comments.

Preparing Revision Guidelines

Sincerely,

Jonathan Eisen

Editor, mSystems

Journals Department
Reviewer comments:

Reviewer #3 (Comments for the Author):

This is a useful resource for the community. There are a few minor issues here and there with the manuscript but one significant one which I detail first below: :

Major issue:

Although this is presented as a manuscript describe a new resource there are some new findings reported here and I believe they make this partly a research paper. Yet they results are not presented in a sufficient way to justify them as research. I would suggest the authors either need to remove new findings OR beef them up better as though they are actual research.

Examples:

* discussion of MLVA markers and which are good. I do not think there is enough herein terms of results to confirm which are effective markers. So I would recommend either removing this from the paper or, changing the terminology to say something like "we computed a score for the predicted utility of the marker based on a computation" rather than calling this "effectiveness" since they are predicting it. Or they need to do more to show evidence that this method works.

* MST typing - same issue as above - there are results presented here on saucer sequences and primers which could be useful to people using the DB but the validity of the analysis is not really documented.

* plasmid typing - same issue as above

If the authors simply implemented known typing tools and referenced them, this would be fine. But there appear to be multiple new types of analysis reported here and those need to be presented in more detail and tested or removed / modified.

Minor comments:

* Could use information on how often this will be updated (more than just saying "periodically")

* Could use information on how long they plan to keep this running

* It would be better to not use Excel for Supplemental material and to instead make things as tsp or csv files or such so they can be used by other programs

Dear Editor,

we thanks the reviewer for the helpful and productive feedback. Please find a point-to-point response below

Major issue: Although this is presented as a manuscript describe a new resource there are some new findings reported here and I believe they make this partly a research paper. Yet they results are not presented in a sufficient way to justify them as research. I would suggest the authors either need to remove new findings OR beef them up better as though they are actual research.

Response to Reviewers: We thank the reviewer for pointing this out. As suggested, we decided to focus the scope of the manuscript to the core functionality and removed the additional results that were listed in detail below by the reviewer.

Examples: discussion of MLVA markers and which are good. I do not think there is enough herein terms of results to confirm which are effective markers. So I would recommend either removing this from the paper or, changing the terminology to say something like "we computed a score for the predicted utility of the marker based on a computation" rather than calling this "effectiveness" since they are predicting it. Or they need to do more to show evidence that this method works.

Response to Reviewers: As proposed we have removed description of this findings.

MST typing - same issue as above - there are results presented here on saucer sequences and primers which could be useful to people using the DB but the validity of the analysis is not really documented.

Response to Reviewers: We thank the reviewer for the raising this point and as suggested we have removed the description of the analysis.

plasmid typing - same issue as above

Response to Reviewers: We have also removed this section.

If the authors simply implemented known typing tools and referenced them, this would be fine. But there appear to be multiple new types of analysis reported here and those need to be presented in more detail and tested or removed / modified.

Response to Reviewers: We appreciate the insight shared by the reviewer, We decided to remove the details of this analysis from the manuscript.

Minor comments: Could use information on how often this will be updated (more than just saying "periodically")

Response to Reviewers: We thank the reviewer for this observation and have now added the update schedule to the manuscript. The database will be updated with obtained new user submissions on a monthly basis.

Could use information on how Long they plan to keep this running

Response to Reviewers: We thank the reviewer for raising this important issue. This platform has several collaborators across board with interest in preservation and sustaining this platform. ZB MED – Information Centre for Life Sciences is an infrastructure and research centre and we have dedicated staff for the maintenance of services; The Bundeswehr Institute for Microbiology is one of the foremost laboratories for *Coxiella burnetii* and due to the occasionally sporadic outbreaks of Q fever, we have a strong motivation to keep the resource available and updated. Due to

this we have no ending of the service planned. Finally, the platform is running on de.NBI infrastructure which is public and government funded for projects across Germany and this platform was built on common and widely used open source tools/libraries hence can be easily taken extended by new software developers.

It would be better to not use Excel for Supplemental material and to instead make things as tsp or csv files or such so they can be used by other programs

Response to Reviewers: We thank the reviewer for this suggestion. We have converted the supplemental materials into csv files.

Kind regards,

Konrad Förstner (for the authors)

October 27, 2021

Dr. Konrad U Förstner
ZB MED - Information Centre for Life Science
Data Science and Services
Cologne
Germany

Re: mSystems00403-21R1 (CoxBase: an online platform for epidemiological surveillance, visualization, analysis and typing of *Coxiella burnetii* genomic sequences.)

Dear Dr. Konrad U Förstner:

I believe the revised manuscript has addressed all the reviewer's concerns quite well and is suitable for publication.

Your manuscript has been accepted, and I am forwarding it to the ASM Journals Department for publication. For your reference, ASM Journals' address is given below. Before it can be scheduled for publication, your manuscript will be checked by the mSystems senior production editor, Ellie Ghatineh, to make sure that all elements meet the technical requirements for publication. She will contact you if anything needs to be revised before copyediting and production can begin. Otherwise, you will be notified when your proofs are ready to be viewed.

As an open-access publication, mSystems receives no financial support from paid subscriptions and depends on authors' prompt payment of publication fees as soon as their articles are accepted. =

Publication Fees:

We recognize that the video files can become quite large, and so to avoid quality loss ASM suggests sending the video file via <https://www.wetransfer.com/>. When you have a final version of the video and the still ready to share, please send it to Ellie Ghatineh at eghatineh@asmusa.org.

Sincerely,

Jonathan Eisen
Editor, mSystems

Journals Department
- 3: Accept
- 1: Accept
- 2: Accept
- 4: Accept